# Recent Advances in High-Content Imaging and Analysis in iPSC-Based Modelling of Neurodegenerative Diseases

**DOI:** 10.3390/ijms241914689

**Published:** 2023-09-28

**Authors:** Giovanna Menduti, Marina Boido

**Affiliations:** Department of Neuroscience “Rita Levi Montalcini”, Neuroscience Institute Cavalieri Ottolenghi, University of Turin, Regione Gonzole 10, Orbassano, 10043 Turin, TO, Italy; marina.boido@unito.it

**Keywords:** iPSC, HCI, personalized medicine, imaging software, data analytics

## Abstract

In the field of neurodegenerative pathologies, the platforms for disease modelling based on patient-derived induced pluripotent stem cells (iPSCs) represent a valuable molecular diagnostic/prognostic tool. Indeed, they paved the way for the in vitro recapitulation of the pathological mechanisms underlying neurodegeneration and for characterizing the molecular heterogeneity of disease manifestations, also enabling drug screening approaches for new therapeutic candidates. A major challenge is related to the choice and optimization of the morpho-functional study designs in human iPSC-derived neurons to deeply detail the cell phenotypes as markers of neurodegeneration. In recent years, the specific combination of high-throughput screening with subcellular resolution microscopy for cell-based high-content imaging (HCI) screening allowed in-depth analyses of cell morphology and neurite trafficking in iPSC-derived neuronal cells by using specific cutting-edge microscopes and automated computational assays. The present work aims to describe the main recent protocols and advances achieved with the HCI analysis in iPSC-based modelling of neurodegenerative diseases, highlighting technical and bioinformatics tips and tricks for further uses and research. To this end, microscopy requirements and the latest computational pipelines to analyze imaging data will be explored, while also providing an overview of the available open-source high-throughput automated platforms.

## 1. Introduction

Experimental use of human induced pluripotent stem cells (hiPSCs) is a powerful research tool to overcome the limitations of human embryonic stem cells or murine primary neurons, and to allow analysis of molecular pathways in live human neurons [1,2,3]. As first described by Yamanaka et al. in 2006 [4], this cellular model finds its milestone in the reprogramming of adult human somatic cells (generally skin fibroblasts or peripheral blood mononuclear cells) into pluripotent stem cells by means of a cocktail of transcription factors. In this way, the generated cellular system preserves human differentiation potentials and genotypes of embryonic stem cells while overcoming ethical concerns [3].

In recent years, iPSC technology has vastly improved, providing more efficient and safe reprogramming techniques and numerous feasible protocols to differentiate iPSCs into a large variety of cell lines and phenotypes [e.g., different neuronal phenotypes such as cortical, dopaminergic, striatal, and motor neurons (MN)] [1]. Furthermore, the increased availability and maintenance advances of iPSC cultures derived from human patients carrying disease-associated mutations [2] offer the advantage, in the context of disease modelling, of avoiding biases in the reproduction of multifactorial and multigenic characteristics of human disorders, compared to primary cultures or the expansion of neuron-like immortalized cell lines [1,2,3]. In the study of neurodegenerative diseases, the use of patient-derived iPSC-based disease modelling platforms as a molecular diagnostic/prognostic tool has already improved the in vitro recapitulation of the pathological mechanisms underlying neurodegeneration. Indeed, this approach has paved the way for characterizing the molecular heterogeneity of disease manifestations, predicting phenotypes that will arise in patients, and enabling low- and high-throughput screening of drug candidates for new therapies (for extended review, see [2,3,5]). However, several technical challenges remain to exploit iPSC-based neurodegenerative disease neuron cultures to improve the scalability, reproducibility, and quality of this type of descriptive disease modelling [6].

A major challenge is related to the choice and optimization of the morpho-functional study designs in patient iPSC-derived neurons, to deeply detail the phenotypes (e.g., damage to neuronal morphology, subcellular organelle dysfunctions, and cytopathies) as markers of late neurodegeneration in vivo [3,5,7]. To date, high-throughput screening assays based on advanced technologies (including robotics, liquid handling, high-sensitivity detectors, and high-performance computing) have enabled full automation in micro-quantitative experiments with large-scale data analysis [8,9]. In this context, the specific combination of high-throughput screening with subcellular resolution microscopy, which provides cell-based high-content imaging (HCI) screening, represents a powerful tool also used for stem cell research and drug discovery. In detail, HCI approaches enable studies of complex heterogeneous biological systems (as mixed cultures) in a high-throughput manner: they provide quantitative observations of comprehensive phenotypes at the subcellular level with spatial and temporal resolution of multiple targets and using multiple measurements as readouts (unattainable goal with traditional approaches) (for a more detailed description, see [10,11,12,13]).

In recent years, HCI analysis methods have enabled in-depth analyses of cell morphology and neurite trafficking in iPSC-derived neuronal cells by using specific microscopes and automated computational assays for several phenotypic screenings (including cell migration, differentiation, and neurite outgrowth) in large datasets from fluorescence microscopy imaging. For instance, these approaches are preferred and required for screening highly complex neurite structures over small-scale screenings with manual or semi-automated quantification of neurite outgrowth, which are subject to time-consuming protocols and possible user-related bias [14,15]. Recent HCI analysis of iPSC-derived neurons have led to enabling in-depth studies of several hallmarks related to neurodegeneration, classifiable to the following: (i) neuronal dysmorphogenesis and survival; (ii) aberrant neuronal protein aggregation and intracellular transport; (iii) mitochondrial dysfunctions; and (iv) compound-induced neurotoxicity (Figure 1). Moreover, several HCI protocols and different cell analysis systems, also useful for drug screening studies, can be employed and classified on the basis of their degree of automation and for their accessibility (i.e., licensed or open-source software; for an extended overview see [14]).

Thus, the purpose of this review is to describe the main recent protocols and the advances achieved with the HCI analysis in iPSC-based modelling of neurodegenerative diseases, highlighting technical and bioinformatics tips and tricks for further uses and research. To this end, microscopy requirements (allowing for imaging of fixed or live cell samples) and the latest computational pipelines for analyzing imaging data will be explored, while also providing an overview of the available open-source high-throughput automated platforms.

### Most Common Microscope Settings and Platform Analysis in iPSC-Based Neuronal Models

To maximize data acquisition in morphological analysis of iPSC-derived neurons, several recent HCI approaches exploit high-throughput microplate imager microscopes [16]. These imaging platforms are suitable for HCI in 24/96/384-well plates and are used in different steps of the experimental design, according to microscope features. For instance, optimization of long-term culture conditions and real-time kinetic information on neuronal cell status and morphological differentiation is feasible through live cell imaging systems for automated live acquisitions and measurements. These microplate imagers, usually equipped with a dedicated suite of assays, automatically collect large datasets of real-time images, in phase contrast and fluorescent modes, from wells stored in the cell incubator [17].

Higher resolution analyses often rely on the use of confocal microscopes with a spinning disk design, also using an increased pinhole-to-pinhole distance, a complementary metal oxide semiconductor scientific camera, and high numerical aperture water immersion objectives: these characteristics assure high-resolution fast imaging of complex cell models, also reducing spectral cross talk [10,18,19,20]. This kind of microscope generally offers the possibility, along with the fastest high-efficiency large-scale imaging of fixed cells in their culture plates, of exploiting the relative specialized licensed software, in order to automatically analyze data with multiple parameters. However, in addition to the licensed image analysis software, HCI analysis for neurite outgrowth and trafficking quantification can be also performed on raw image files exported after acquisition by using a wide variety of open-source software with comparable workflow pipelines (for a detailed description of the employed microscopes and analysis platforms, see Table 1 and Table 2) [21,22,23,24]. Recently, HCI analysis workflows have also been implemented and improved with the development of artificial intelligence (AI) technology. Indeed, among its various applications, AI technology is specialized for image recognition, which is already being exploited in HCI assessment of iPSC-derived cell disease-specific phenotypes and drug screening, from label-free microscopic images. Hence, different programs are also leveraged for HCI specialized AI. For example, machine learning technology, which enables computer systems to learn and predict response from unknown datasets according to a pre-trained program, is the most prominent one used for AI, often, for example, in neural network analysis [23,25,26,27].

Finally, since conventional confocal microscopy provides a resolution spectrum limited to hundreds of nm, in order to overcome the diffraction limit (e.g., the size of synaptic vesicles is about 40 nm), the HCI analysis-based experiments can currently benefit from the use of super-resolution methods, such as structured illumination microscopy (SIM), stimulated emission exhaustion microscopy (STED), or photoactivated localization microscopy (PALM). In fact, these applications have recently reached neuronal investigations focused on the characterization of the spatial organization of proteins in synapses, of the structures of the axon cytoskeleton, or of the temporal dynamics of vesicle fusion [28,29,30].

**Table 1 ijms-24-14689-t001:** Overview of open-source software and related workflow pipelines for iPSC-based HCI analysis.

Open-Source Analysis Software Advantages: Open-Source, Enabling Single-Cell Tracing and Measurements, Free Plug-Ins Download for Several Cellular Analysis Types, Feasible for Custom-Made Algorithm Analysis or Macros				
Analysis Platform Software	HCI Analysis	Plug-In and Tools	Microscopes (Used in the Mentioned References)	References
CellProfiler (automated) (https://cellprofiler.org/, accessed on 24 September 2023)	-neurite outgrowth	-Software analysis pipeline: https://doi.org/10.5281/zenodo.6642365 (accessed on 24 September 2023)	-Operetta CLS High-Content Analysis System—with Harmony software (PerkinElmer, Waltham, U.S.)	-CellProfiler software: [31]; -[32]
	-mitochondrial fitness, neuronal toxicity quantification of neuronal branching complexity	-Software analysis pipeline: https://github.com/StemCellMetab/Mitochondrial-membranepotential (accessed on 24 September 2023)	-Operetta CLS High-Content Analysis System with Harmony software (PerkinElmer, Waltham, U.S.)	-[33,34]
	-mitochondrial function, morphology and cell viability	-Software analysis pipeline: automated synaptic imaging assay (ASIA), https://github.com/thayerlab/ASIA-pipelines scripts written (accessed on 24 September 2023)	-Opera High-Content Screening System, (live imaging) (PerkinElmer, Waltham, U.S.)	-[35]
	-discrimination in synaptic density changes		-Nikon, Tokyo, Japan A1 confocal microscope (Nikon, Tokyo, Japan)	-[36]
ImageJ (semi-automated) (https://imagej.nih.gov/ij/download.html, accessed on 24 September 2023)	-Neurite outgrowth, growth cone, axonal swellings	-ImageJ-NeuronJ plug-in -ImageJ-Neurite tracer macro	-Axioplan2 (Carl Zeiss AG, Oberkochen, Germany), LSM-710 (Carl Zeiss AG, Oberkochen, Germany), BZ9000 (Keyence, Itasca, U.S.), or IN Cell Analyzer 6000 (GE Healthcare, Chicago, U.S.)	-ImageJ software: [24,37]; -NeuronJ plug-in: [38]; -Neurite tracer macro: [39]; -[40]
	-Neurite outgrowth, axon degeneration index; protein aggregates automated quantification	-ROI manager tool, Threshold function, analyze particles plug-in; cell counter plug-in -Image Mining: custom-made image processing and analysis application with an extendable “plug-in” infrastructure (based on data mining, AI, machine learning, image retrieval, image processing, computer vision and database)	-Opera High-Content Screening System (PerkinElmer, Waltham, U.S.)	-Axon degeneration index: [41,42,43]; -Image mining: [44]; -[26]
	-Motility of fluorescently labeled organelle and neurite number quantification	-Pairwise Stitching plug-in; Simple Neurite Tracer plug-in with Sholl Analysis; -segmented line and ROI manger tool; -Multiple Kymograph plug-in; -Custom MATLAB GUI (Kymograph Suite) (Manually tracing of individual organelles)	-UltraView Vox Spinning Disk Confocal system (PerkinElmer, Waltham, U.S.) with a Nikon Eclipse Ti inverted microscope (Nikon, Tokyo, Japan); inverted DMI6000B microscope (Leica Microsystems, Wetzlar, Germany) using LAS-X software (Leica Microsystems, Wetzlar, Germany).	-Pairwise Stitching: [45]; -Sholl Analysis: [46,47]; -[48]
	-Membrane trafficking	-Reslice function (Kymograph construction) (https://imagej.nih.gov/ij/plugins/radial-reslice/index.html, accessed on 24 September 2023)	-Incucyte SX1 live-cell analysis system (Sartorius, Göttingen, Germany); Nikon, Tokyo, Japan Eclipse Ti microscope (with optical autofocus system and a motorized piezo stage) spinning disk microscope (Nikon, Tokyo, Japan) (real-time quantitative live imaging); -Andor Ixon Ultra (EM-CCD) camera and the MetaMorph software imaging system (Molecular Devices, San Jose, U.S.);	-[49]
	-Neuronal local neuronal secretory system	-Custom-made macro intracellular for quantification of intracellular markers colocalization (%)	-Leica SP8 confocal microscope and a LASX imaging system (Leica Microsystems, Wetzlar, Germany).	-[50]
	-Discrimination in synaptic density changes	-Software analysis pipeline:automated synaptic imaging assay (ASIA), https://github.com/thayerlab/ASIA-pipelines scripts written (accessed on 24 September 2023)	-Nikon Eclipse Ti-E inverted confocal microscope and the NIS Element software (Nikon, Tokyo, Japan) + Carl Zeiss LSM 880 AiryScan confocal microscope and the Zen Black 2.3 software, within the AiryScan super-resolution mode (Carl Zeiss AG, Oberkochen, Germany)	-[36]
	-Axonal outgrowth and muscle maturation	-ImageJ macro for calculating pillar deflection: Method A: Supplementary Data 4 of [15]. Method B: Supplementary Data 5 of [15].	-Nikon A1 confocal microscope controlled with the JOBS module of Nikon Elements software (Nikon, Tokyo, Japan) -Zeiss, Axiovert 200 (Phase-contrast) (Carl Zeiss AG, Oberkochen, Germany); -Olympus, model no. FV-1000 (Confocal laser microscope with motorized stage) (Olympus, Tokyo, Japan) Tokai Hit, INUG2F-ZM (Tokai Hit, Fujinomiya, Japan) (Phase-contrast and fluorescent microscope with a stage-top incubator)	-[15]
	-Autophagy LC3-based assay	-Customed R script (https://www.r-project.org/; accessed on 24 September 2023) version 3.5.2 and data processing with Bioconductor R package cellHTS2 (https://www.bioconductor.org/packages//2.7/bioc/html/cellHTS2.html, accessed on 24 September 2023) Coloc2 plug-in for Fiji (providing Pearson’s R correlation) (https://imagej.net/Coloc2; accessed on 24 September 2023);	-Opera Phenix High-Content screening System with Harmony software (PerkinElmer, Waltham, U.S.)	-[51]
	-Intracellular transport	-plusTipTracker software (for microtubule dynamics video quantification)	-Olympus Inverted FV1000 confocal microscope (Olympus, Tokyo, Japan); -STED imaging was performed on a custom built, dual color, beam scanning system; -Leica SP5 microscope equipped with a controlled environment chamber (Leica Microsystems, Wetzlar, Germany).	-plusTipTracker: [52]; -[53]

**Table 2 ijms-24-14689-t002:** Overview of licensed software and related workflow pipelines for iPSC-based HCI analysis.

Licensed Analysis Software Advantages: Allowed with Licensed Microscopes, Powerful Image Analysis Capabilities with Highly Flexible and Easy-to-Use Building Blocks to Analyze Simple and Complex Phenotypes of Cells, Automated Cell Tracking, Automated Multiple Segmentation and Co-Localization Analysis, Fast Automated Cell Analysis (Minutes) Enabling Multi-Threaded, Parallel Image Processing, Teachable Interface for Analysis Creation, and Batch Processing for Large Time-Lapse Image Datasets.				
Analysis Platform Software	HCI Analysis	Building Blocks for Analysis Segmentation and Tools	Required Microscopes	References
-Harmony High-Content Imaging and Analysis Software(PerkinElmer, Waltham, U.S.) (https://www.perkinelmer.com/it/product/harmony-4-8-office-hh17000001, accessed on 24 September 2023)	-Neurite outgrowth and neuron maturation assessment	-Find nuclei, Find neurites, Calculate Intensity Properties	-Opera PhenixPlus CLS High-Content screening System (PerkinElmer, Waltham, U.S.) with CSIRO Neurite analysis software (https://www.csiro.au/en/research/technology-space/data/neurite-analysis-software, accessed on 24 September 2023)	-[54]
	-Intracellular protein aggregation	-Find Spot	-Operetta or Opera Phenix CLS High-Content Analysis System (PerkinElmer, Waltham, U.S.)	-[55]
	-neurite outgrowth	-nuclear parameters neurite parameters	-Opera CLS High-Content Analysis System (PerkinElmer, Waltham, U.S.)	-[56]
-Columbus (image data storage and analysis system allowed for connection with Harmony software) (PerkinElmer, Waltham, U.S.) (https://www.perkinelmer.com/it/product/harmony-4-8-office-hh17000001, accessed on 24 September 2023)	-Mitochondrial fitness and neuronal toxicity and quantification of neuronal branching complexity	-Software analysis pipeline: https://github.com/StemCellMetab/Mitochondrial-membrane-potential (accessed on 24 September 2023)	-Operetta CLS High-Content Analysis System (PerkinElmer, Waltham, U.S.	-[33,34]
	-Autophagy LC3-based assay		-Opera Phenix CLS High-Content screening System (PerkinElmer, Waltham, U.S.)	-[51]
-MetaMorph Microscopy Automation and Image Analysis Software (Molecular Devices, San Jose, U.S.) Automated (https://www.moleculardevices.com/products/cellular-imaging-systems/acquisition-and-analysis-software/metamorph-microscopy, accessed on 24 September 2023)	-Neurite outgrowth	-Software analysis: https://www.moleculardevices.com/applications/neurite-outgrowth (accessed on 24 September 2023)	-MetaMorph Microscopy Automation and Image Analysis Software (Molecular Devices, San Jose, U.S.)	-[57]
	-Membrane trafficking		-MetaMorph Microscopy Automation and Image Analysis Software (Molecular Devices, San Jose, U.S.)	-[49]
-IN Cell Analyzer 6000 software (GE Healthcare, Chicago, U.S.)	-Neurite outgrowth	-Segmentation of ROI (Dendrites, cell bodies, and axons.)	-IN Cell Analyzer 6000 high-performance and high-content automated laser-based confocal imaging platform and ImageXpress Micro Confocal High-Content Imaging System (GE Healthcare, Chicago, U.S.)	-[58]
	-Intracellular protein aggregation		-IN Cell Analyzer 6000 IN Cell Developer Toolbox version 1.9 (GE Healthcare, Chicago, U.S.)	-[59]
	-Cell population assays, fluorescence intensity analysis, neurite length analysis		-IN Cell Analyzer 6000 IN Cell Developer Toolbox version 1.9 (GE Healthcare, Chicago, U.S.)	-[60]
	-Neuronal classification and outgrowth convolutional neural network analysis (random forest classification using total neurite length, number of cells, and average size of neuronal soma as random classifiers)	-* Keras/TensorFlow framework (v1.13.1)12 on GTX1080Ti by using CUDA 10.0. scikit-learn (v0.23.2), gradient-weighted class activation mapping (Grad-CAM) and guided Grad-CAM algorithm	-IN Cell Analyzer 6000 high-performance and high-content automated laser-based confocal imaging platform (GE Healthcare, Chicago, U.S.)	-[61,62]; -[27]
	-Image segmentation in individual mitochondria (masking of somatic, axonal, and dendritic mitochondria)	-Cell bodies count and analysis of the number, area, median circularity, and length of mitochondria	-IN Cell Analyzer 6000 confocal microscope (GE Healthcare, Chicago, U.S.) and GE Developer Toolbox (1.9.2, build 2415) software (GE Healthcare, Chicago, U.S.)	-[63,64]
-CL-Quant Automated Image Analysis Software (Nikon, Tokyo, Japan) (https://www.nikon.com/company/news/2019/1008_cl-quant_01.html, accessed on 24 September 2023)	-Cell population assays, fluorescence intensity analysis, neurite length analysis	-Nuclei and neurite tracing, fiber objects quantification (neurite lengths)	-BioStation CT (Nikon, Tokyo, Japan)	-[60]
-Cellomics software (Thermo Fisher Scientific, Waltham, U.S.) (https://www.thermofisher.com/it/en/home/brands/thermo-scientific/cellomics.html, accessed on 24 September 2023)	-Cell population assays, fluorescence intensity analysis, neurite length analysis	-Nuclei and neurite tracing, fiber objects quantification (neurite lengths)	-ArrayScan high-content system (Thermo Fisher Scientific, Waltham, U.S.)	-[65]
-Imaris (Bitplane, Belfast, UK) (Not requiring specific HCI microscope) (https://www.oxinst.com/search-results?search=IMARIS&businesses=bitplane, accessed on 24 September 2023)	-Membrane trafficking	-Surface function 3D cellular structures reconstruction from different image dataset	-Andor Ixon Ultra (EM-CCD) camera and the MetaMorph software (Molecular Devices, San Jose, U.S.) imaging system Leica SP8 confocal microscope and a LASX imaging system (Leica Microsystems, Wetzlar, Germany)	-[49]
	-Axonal outgrowth and muscle maturation		-Zeiss, Axiovert 200 (Phase-contrast) (Carl Zeiss AG, Ober-kochen, Germany); Olympus, model no. FV-1000 (Confocal laser microscope with motorized stage) (Olympus, Tokyo, Japan) with a stage-top incubator (Tokai Hit, INUG2F-ZM, Tokai Hit, Fujinomiya, Japan) (Phase-contrast and fluorescent microscope)	-[15]

* Platform and plug-ins available for trained AI or machine-learning driven data analysis and classification.

## 2. HCI Analysis of Neuronal Dysmorphogenesis in iPSC-Based Neurodegenerative Diseases Modelling

Nowadays, phenotypic analysis of human iPSC (hiPSC)-based disease models is the most reliable probe (i) to define cellular differences and vulnerabilities in patients compared to healthy control cells, as well as (ii) to investigate the underlying molecular mechanisms of various neurodegenerative processes [56,66,67,68]. Cell body shrinkage, axonal degeneration, and/or neurite pathologies are key pathological hallmarks in a number of neurodegenerative diseases, including spinal muscular atrophy (SMA), amyotrophic lateral sclerosis (ALS), Alzheimer’s disease (AD), Parkinson’s disease (PD), and hereditary spastic paraplegia (HSP) [14,40]. Outlining and quantifying the degree of aberrant neuronal morphology, which strongly affects the connectivity and the processing/distribution of information within neural circuits [14], is a critical step to apply to patient-derived iPSC analysis for identifying early disease-specific defects and designing possible therapeutic strategies. Once the specific immunohistochemical or immunofluorescent reactions have been performed, the evaluation of neurite/soma morphology generally requires a multiparametric analysis, including neurite length and number; roots (i.e., the emerging neurites from the cell body; branches and nodes (i.e., the ramification points); internodal segments; and tips and soma sizes (Figure 2). This analysis allows for quantification of neuron maturation, synaptic loss, and/or atrophy and degeneration.

### 2.1. Experimental Design and Analysis Setting for hiPSC-Based HCI for Neuronal Morphology Phenotypes

In recent years, several HCI analysis methods have been published for in-depth studying of the morphological changes in iPSC-derived neuronal cells. To this aim, suggestions for specific microscope setups and choice of automated computational assays for a high-throughput skeletonization approach in large datasets from fluorescence microscopy imaging have been provided (Figure 2). Here we report some examples of experimental workflows and related applications in patient iPSC-based models of neurodegeneration. All the details about the employed microscopes and analysis platforms are listed in Table 1 and Table 2.

The works of Lickfett and colleagues in 2022 provide a step-by-step description on how to perform HCI imaging for quantification of both axonal and dendritic length, number of branch points, staining signal areas, and other parameters in 384-well plates of fixed iPSC-derived neurons [32]. Specifically, the cells of interest were obtained through various differentiation approaches (NGN2 induction [69] from the engineered iPSC line BIHi005-A-24, dopaminergic neurons obtained by exposing iPSCs to specific growth factors and small molecules) [32]. The protocol-specific settings have been designed for use in a high-content analysis (HCA) system spinning disk microscope (in both confocal and non-confocal imaging mode) equipped with relative licensed imaging analysis software. The work also provides an example of HCI analysis for the quantification of neurite outgrowth to be performed using the open-source software CellProfiler 4.2.1 [31] as a robust tool that enables automated image processing from high-throughput experiments image datasets: it allows the analysis of neurite outgrowth and branching of differentiated iPSC-derived neurons to eventually unravel disease-specific morphological defects.

Another similar example was provided by Wali et al. in 2023: the authors performed a high-content screening to measure neurite outgrowth and length, roots, extremities, and segments in immature (Day 1 post seeding neural progenitor cells) and mature neurons (Day 15) [54] by using a confocal HCA system spinning disk microscope. In particular, the neuronal morphology HCI analysis was performed on 96 well-plated fixed neurons through the licensed Harmony software 5.1 (PerkinElmer, Waltham, U.S.). The software building blocks to image segmentation enabled breaking down the image into discrete objects (such as individual nuclei or neurites), detected according to immunostaining of specific markers. Specifically, the analyses were performed by first segmenting the image using the “Find Nuclei” building block: this tool identifies each nucleus as reference point for a cell object from which the MAP2-positive neurites extend, thus continuing the segmentation of the neurites with “Find Neurites” building block exploitation and the measurement of the related outgrowth parameters. Notably, these segmentation tools for evaluating neuron morphology could otherwise be obtained from other licensed image analysis software (such as MetaXpress software, Molecular Devices, San Jose, U.S., https://www.moleculardevices.com/products/cellular-imaging-systems/acquisition-and-analysis-software/metaxpress, accessed on 24 September 2023) and from open-source image analysis software such as the above-mentioned Cell Profiler and ImageJ (https://imagej.nih.gov/ij/download.html, accessed on 24 September 2023) [37] with the NeuronJ plugin [38].

The limitations of the mentioned protocols are related to the automated imaging approaches of cultured neurons: for instance, automated analysis might not involve single cell assessment of dendritic arborization (possible to perform with Sholl analysis [46,47]) or might not avoid possible axonal/dendritic crossing-over from two or more neighboring neurons. Although new plug-ins are not excluded from being implemented in the available HCA pipelines, careful experimental design is generally recommended (setting the optimal seeding density and fixation time point for each neuronal cell type) [32,70,71].

### 2.2. HCI Analysis of Neuronal Morphology Phenotypes in Patient iPSC-Based Neurodegeneration Modelling

Nowadays, modelling neurodegenerative diseases by HCI analysis in iPSC-derived neurons is common in different research fields, such as AD, PD, and MNs diseases.

For example, Chang et al. (2019) exploited the HCI of neuronal morphology as a critical step in the experimental characterization process of AD-relevant cellular features in an iPSC model generated from two familial AD patients carrying a mutation heterozygous D678H in the *APP* gene (AD-iPSC) [57]. Herein, the neurite outgrowth measurements of differentiated neurons (including total outgrowth, processes, and branches) were assessed, following HCA-microscope system spinning disk microscope cell images acquisition, by MetaMorph microscopy automation image analysis software (Molecular Devices, San Jose, U.S.), detecting the neurites by the immunofluorescence staining of the neuronal marker TUBB3. The work highlighted the reduced neurite outgrowth in AD-iPSC-derived neurons, in association with aberrant accumulation of Aβ and tau phosphorylation, compared to wild-type iPSCs [57].

Bassil et al. (2021) performed a similar experimental analysis and applied a comparable workflow pipeline in a complex multicellular system. Indeed, they generated an automated, consistent, and long-term culturing platform of hiPSC AD neurons, astrocytes, and microglia to investigate and quantify Aβ plaque-induced dystrophic neurites, synapse loss, dendrite retraction, axon fragmentation, and neuronal cell death [58]. The workflow began with induced differentiation of iPSC-derived neurons in large batches, which were then replated in 384-well imaging plates, with following serial addition of differentiated astrocytes and microglial cells. Automated confocal image acquisition and analysis were performed with a HCA-microscope system spinning disk microscope and with customed imaging analysis software. Herein, 384-well plates enabled simultaneous testing of numerous experimental conditions: image analysis scripts indeed provided precise segmentation of markers of interest, including dendrites (MAP2), cell bodies (CUX2), axons [Tau, phospho-tau (p-Tau)], and synapses (Synapsin 1/2), and thus measures of related outgrowth. The results of the work provided an in vitro recapitulation of the hallmarks of AD, developed in a sequential order of events comparable to the disease progression of AD in humans. In particular, HCA imaging of neuronal cells exposed to chronic treatment with soluble Aβ42 [72] allowed the accurate reconstruction and quantification of the degree of p-Tau-positive dystrophic neurite formation around the plaques, microtubule fragmentation and axon degeneration, and consequent loss of nuclear CUX2 and dendrite atrophy in different stages of the disease [58].

The same above-mentioned HCI automated analysis tools and methods were successfully applied on hiPSC-MNs to study their morphology for MN diseases (MNDs) modelling. In particular, the Fujimori (2018) and Imamura groups (2021) performed a phenotypic discrimination in the pattern of neuronal degeneration using a large sampling of hiPSCs from ALS and healthy controls. In more detail, relying on parameters of total neurite length, cell number, and mean neuronal soma size, the two works obtained phenotypic discrimination respectively between several heterogeneous sporadic ALS iPSC models [60] and between images derived from healthy control subjects or ALS patients in constructing an AI-based predictive model of ALS [27,61,62].

As a further example of HCI in iPSC-based models of MNDs, Rehbach et al. (2019) performed a multiparametric screening in hereditary spastic paraplegia (HSP) patient-specific neurons. They reported the synergistic exploitation of both licensed and open-source imaging software to achieve shorter read-out times in the quantification of HSP-induced neuronal morphology degeneration from different phenotypic assays [40]. The study analysis indeed, following 96-well plated fixed-cells acquisition with different HCA confocal spinning disk microscopes, relied on differentiated imaging analysis: (i) neurite outgrowth assays were analyzed (24 h after plating), first in a semi-automated manner by using ImageJ and the NeuronJ plug-ins [21] and then in automation mode with the InCell Developer licensed software toolbox (GE Healthcare, Chicago, U.S.); (ii) concomitant growth cone assays were conducted using a semi-automated CellProfiler script [31] or an automated InCell Developer script; (ii) axonal swelling assays quantification was performed manually (5 days after plating) and normalized to the length of TAU1 positive axons as determined via ImageJ with the Neurite Tracer macro [39] or the InCell Developer toolbox. The designed multiparametric and rapid phenotyping approach highlighted that HSP-iPSC neurons have a 51% reduction in the growth of neurites and a 60% increase in growth cone area and axon swellings (after only 5 days), suggesting its feasible application also to other iPSC-based research models of neurodegenerative diseases.

## 3. HCI Analysis for Aberrant Neuronal Protein Aggregation and Intracellular Transport in iPSC-Based Neurodegenerative Disease Models

The major factors contributing to the collapse of the synaptic network structure and function (leading to neurodegeneration) include defective long-range pathological protein aggregation, intracellular transport, aberrant proteostasis, cytoskeleton abnormalities, impaired mitochondrial homeostasis, DNA and RNA defects, and neuroinflammation [73]. Recent works have contributed to clarify the interrelationships of the cellular and molecular processes underlying neurodegeneration and how hallmarks can be detected and monitored using HCI analysis methods (Figure 3). For instance, several approaches based on the feasibility of gene editing (by CRISPR/Cas9), transfection and transient expression of genetically encoded fluorescent markers, or the employment of advanced fluorescent probes have favored the HCI of neuronal and organelle protein dynamics and their tracking by automated analysis [48,73].

Here, we report a few examples of HCI approaches and automated analysis tools for live or fixed-samples imaging that enabled the delineation of neurodegeneration-related molecular hallmarks in human iPSC-derived neurons. A detailed description of the employed microscopes and analysis platforms is provided in Table 1 and Table 2.

### 3.1. HCI Analysis for Aberrant Neuronal Protein Aggregation in iPSC-Based Neurodegenerative Disease Models

Aberrant protein aggregation is a key pathological hallmark and diagnostic marker of a variety of proteinopathies, including AD, PD, and ALS, respectively characterized by abnormal folding and aggregation of Aβ peptides [58], αSyn proteins [74], and TAR DNA binding protein 43 (TDP-43) or superoxide dismutase 1 (SOD1) proteins [75,76,77]. In this context, the application of HC imaging analysis in a patient’s iPSCs enables the quantification of intracellular protein aggregates by specific tracing of selected fluorescence markers.

In 2022, Manos et al. provided an example of HCI of protein aggregation aimed at establishing a physiologically relevant AD model for studying the tau pathobiology [55]. In more detail, the work focused on the evaluation of possible endogenously induced protein aggregation in *MAPT* (microtubule-associated protein tau gene) WT and CRISP-CAS9 edited *MAPT* KO iPSC neurons in response to AD brain-derived tau species. With this goal, a timeline for cell seeding and amplification to determine the minimum analysis window to achieve quantifiable Tau aggregation for high-throughput applications was established, further examining multiple time points to determine the impacts of tau aggregation on cell health. HCI of immunofluorescence staining for Tau aggregation was conducted on an HCA-microplate reader microscope fitted with the related automation analysis software for images analysis; tau aggregates were defined using MC1 staining and then analyzed throughout the “Find Spot” segmentation algorithm. Furthermore, for evaluating intracellular localization of tau seeds, confocal microscopy was performed, and MAP2-somatodendritic staining was utilized as a marker of the cell status [55]. Overall, the obtained results demonstrated the increase of insoluble, endogenous tau aggregates in hiPSC-derived cortical neurons seeded with AD brain-derived competent tau species. The work provides a successful demonstration of combining human neurons, endogenous tau expression, and AD brain-derived competent tau species, offering a more physiologically relevant platform to study tau pathobiology or, more in general, to recapitulate protein aggregation mechanisms.

Concerning PD, automated quantification of αSyn neurotoxic protein aggregates in patient iPSCs has been reported in the Antoniou et al. work (2022), in which the aggregation process was detected as a fluorescent signal in PD patient p.A53T-iPSC-derived neurons [26]. To this aim, a red fluorescent molecular rotor dye (PROTEOSTAT Aggresome Detection Kit; Enzo) was administered to the neurons: it becomes brightly fluorescent when it binds to protein aggregates. Then, immunolabeling for TUJ1 or TH22 markers was performed. Finally, manual analysis of protein aggregates from image dataset was performed by isolating individual cells from images selected from the region of interest (ROI), applying a threshold, and utilizing the “analyze particles” ImageJ function [24,37]. Additional details are provided and commented on in Section 4.3.

A similar example comes from the ALS field: indeed, a rapid and robust high-throughput screening system for TDP-43 or SOD1 protein aggregation in ALS iPSC cells has been provided by Kondo et al. (2023) to develop a cell culture system allowing feasible analysis of pathological protein aggregation and hyperexcitation in iPSC-derived MN models of sporadic ALS [59]. Using MNs generated from genome-edited and patient iPSCs carrying a mutation in the fused sarcoma (FUS) and SOD1 genes, and from a healthy patient’s cell line as controls, HCI analysis of immunostained 96-well plates was performed using the HCA microplate reader microscope. Then, quantitative analysis of protein accumulation (based on immunofluorescence marker staining for different protein aggregates) was performed using the licensed HCI automated analysis software IN Cell Developer Toolbox version 1.9 (GE Healthcare, Chicago, U.S.): the sum cytosolic FUS+ aggregate area, G3BP+ aggregate area, FUS + G3BP+ aggregate area, or SOD1+ area/βIII+ cell number were quantified. By reproducing several ALS phenotypes (successfully assessed for their protein accumulation and neuronal death by the robust detailed phenotypic screening), the proposed experimental workflow appears suitable to facilitate the discovery of new therapies for ALS and, more broadly, to favor stratified and personalized medicine for MNDs.

### 3.2. HCI Analysis of Axonal Transport of Endogenously Labelled Vesicles in Neurodegenerative Patient iPSC-Derived Neurons

Neurons rely on intracellular transport to deliver newly synthesized organelles and macromolecules to the far-reaching ends of their neurites and to carry cargos (such as neurotrophic factors) back to the soma to alter gene expression. Defects in long-range intracellular transport have emerged as a common factor of several neurodegenerative disorders, including PD, AD, and MNDs [48,78], and can be unraveled by HCI studies.

A work by Boecker et al. in 2020 described an approach for live HCI analysis of axonal and dendritic transport in iPSC-derived cortical neurons, enabled by monitoring the motility of fluorescently labelled organelle markers, including early, late, and recycling endosomes, as well as autophagosomes and mitochondria [48]. By performing inducing transient expression in iPSC-derived neurons of genetically encoded fluorescent markers to characterize the motility of Rab-positive vesicles [79] and generating, by genome editing, LAMP1-EGFP knock-in iPSCs, the study exploited different image acquisition methods, followed by open source analysis platform exploitation for whole quantifications. In detail, neurons (with a focus on axon imaging) were live-imaged in an environmental chamber at 37 °C on an inverted microscope spinning disk confocal system (for imaging Rab-, LAMP1- and LC3-vesicles, and mitochondria); phase-contrast images of differentiating glutamatergic cortical GFP-labeled neurons for morphology studies were instead recorded with an inverted confocal microscope using LAS-X software (Leica Microsystems, Wetzlar, Germany). To analyze the whole neuron, images were stitched together using the FIJI plug-in Pairwise Stitching [45]. While neurite number quantification was performed on stitched images using the FIJI plug-in, simple neurite tracer was performed with Sholl analysis [41,46,47]. The authors discriminated axons and dendrites based on morphological parameters, including diameter, length, and branching of the extensions [80]. Therefore, axons were manually traced and measured using the FIJI segmented line and ROI manager tools. Finally, kymographs were generated using the Multiple Kymograph plug-in for FIJI, while tracks of individual organelles were manually traced using a custom MATLAB GUI (Kymograph Suite). The work provides a robust approach for live imaging and measurement of axonal and dendritic transport in iPSC-derived cortical neurons by exploiting transient overexpression of organelle markers and generating LAMP1-EGFP knock-in iPSCs to study transport of fluorescently labelled or endogenously labelled vesicles, respectively. These powerful HCI methods could also be extendable to more complex iPSC-derived neural cell systems, allowing for in-depth characterization of organelle dynamics in neurodegenerative disease progress.

In this regard, more recently, the work of Wang et al. (2023), further exploiting the HCI of live-imaged hiPSC-derived neurons, provided an important advance in defining the specialized pathways of membrane trafficking in human neurons [49]. The study shed light on the dynamics of the local dendritic secretory system and on the structural stability of organelles in the neuronal transport of newly synthesized proteins by quantifying the spatial and dynamic behavior of the dendritic Golgi elements and endosomes in neuronal differentiation [81]. A real-time quantitative live-cell imaging system was used both to monitor growth and differentiation of cells and to highlight the distinct dynamics of each type of endosome in the dendrites of human neurons during early neuronal development. To allow fluorescent endosome tracking and subsequent analysis, neurons were loaded with the red BODIPY TR ceramide Golgi staining reagent (Abcam, Cambridge, UK) and/or transduced with various GFP-tagged protein markers (as GFP-tagged wild-type Rab5, Rab7, or Rab11 plasmids). For higher resolution live-cell imaging (enabling the visualization of dynamic endosome movements within the dendrites of mature neurons), an inverted confocal spinning disk microscope (with an optical autofocus system and a motorized piezo stage), an Andor Ixon Ultra (EM-CCD) camera and the MetaMorph software (Molecular Devices, San Jose, U.S.) imaging system were used. Kymographs for dendrite images were constructed using the “Reslice” function in FIJI/ImageJ software (https://imagej.nih.gov/ij/plugins/radial-reslice/index.html, accessed on 24 September 2023). Image acquisition for fixed samples was performed using a confocal microscope and a LASX imaging system. Then the images were sequentially collected for multicolor imaging, enabling tracking of the fluorescent labelling of specific organelle markers in the MAP2+ dendrite: KDEL (ER), GM130 (cis-Golgi), p230/golgin245 (trans-Golgi network [TGN]), EEA1 (early endosome), CD63 (late endosome/lysosome), Rab11 (recycling endosome), and Tom20 (mitochondria). Following the deconvolution of raw microscopy images using Huygens software (Scientific Volume Imaging, Hilversum, Netherlands), the 3D cellular structures from different image datasets were reconstructed using the “Surface” function to measure relevant parameters. Finally, the analysis was performed using Imaris software (Bitplane, Belfast, UK). This study provided a reliable model to study molecular players and pathways regulating dendritic Golgi and local protein trafficking in human neurons, laying the foundation for defining the trafficking pathways associated with neuronal networks in health and disease.

IPSC organelles fluorescent labelling has also been recently applied in the neurodegenerative disease field in order to investigate the neuronal secretory system [50]. This study was intended to better substantiate how the physical interaction of APP protein with β (BACE1)- and γ-secretase in endo-lysosomal organelles affects APP processing in the generation of amyloid Aβ (a central point in AD research) [50]. The experimental evaluations have been carried out on low Aβ-secreting neural progenitor cells and high Aβ-secreting mature hiPSC-derived neurons. In these cells, the colocalization of both full-length APP/BACE1 and the APP-derived C-terminal fragment/presenilin-1 (the catalytic subunit of γ-secretase) were evaluated with immunocytochemistry combined with a proximity ligation assay (PLA) [82]. This assay, taking advantage of short oligonucleotide-tagged antibodies, can highlight fluorescently the proximity of two target proteins. Then colocalization analysis was normalized on the number of cells by calculating the ratio between the area of PLA dots and the area of nuclei. The immunocytochemistry-stained cell analysis was performed using an inverted confocal spinning disk microscope. The analysis of colocalization of APP and the secretases by PLA and that of intracellular localization of PLA dots were performed by using confocal microscopy, within the AiryScan super-resolution mode, to thoroughly evaluate the protein interactions. Of note, image analysis was performed using a custom-made macro for ImageJ [37], for which all detailed processing steps are described in the paper methods [50]. Another custom-made macro for Image J was employed and described for the analysis of intracellular localization of APP/secretase colocalization, calculated with the percentage of organelle (GFP-tagged organelle marker) that was occupied by PLA dots. This work contributed to clarifying the AD biology by studying the regulation of APP-secretase interaction in Aβ-secreting mature human neurons and suggesting a novel therapeutic strategy for this disease.

HCI analysis for in-depth characterization of intracellular transport in iPSC modelling of neurodegenerative diseases also relied on the use of super-resolution methods. In this regard, the work of Paonessa et al. (2019) focused on the effects of Tau gene (*MAPT*) mutations on nucleocytoplasmic transport as a pathological mechanism contributing to frontotemporal dementia (FTD)-induced neuronal degeneration by performing STED-based super-resolution imaging of Tau colocalization in the invaginations of the nuclear lamina [83]. In more detail, the HCI analysis was performed in FTD patient-derived iPSC neurons carrying different *MAPT* mutations with different microscopy methods [53]. Indeed, the mislocalization of hyperphosphorylated tau in the cell bodies and dendrites of FTD-neurons was assessed using confocal microscopy, while the colocalization imaging of tau to the outer nuclear membrane, particularly within hundreds of nanometers of the nuclear lamina, was allowed by STED microscopy (reference settings are indicated in the work methods). For image analysis, several ImageJ software plug-ins were exploited: the colocalization of tau and MAP2 was calculated using the Coloc2 plug-in for Fiji, providing Pearson’s R correlation (https://imagej.net/Coloc2, accessed on 24 September 2023); quantification of nuclear invaginations in neurons was achieved using a custom plug-in for Fiji bioimage analysis software [37] based on nuclear co-staining for LaminB1 and DAPI. Further live imaging analyses of microtubule dynamics were performed on neurons grown in single 35 mm m-Dish dishes (Ibidi) and transfected with a plasmid encoding for GFP-EB3 using a confocal microscope equipped with a controlled environment chamber (37 °C; 5% CO_2_) and allowing resonant scanning acquisition (for real-time imaging). The latter acquisitions (in video form) were analyzed using the plusTipTracker software [52]. The results of the work provided an in-depth assessment of tau mislocalization in the FTD-*MAPT* neuronal cell body, which was shown to underlie abnormal movements of microtubules that deform the nuclear membrane. These data indicate that impaired nucleocytoplasmic transport is a pathogenic mechanism in multiple forms of neurodegenerative disease, and the application of super-resolution microscopy could be widely employed to perform similar analysis in other neurodegenerative models of iPSCs as well.

### 3.3. iPSC-Derived Neurons HCI Analysis of Mitochondrial Dynamics and Homeostasis

Synaptic function requires a tight regulation of mitochondrial function and energy supply to maintain correct neuronal signaling, elimination, and replenishment of constituents for proper synaptic function, as well as general neuronal coordination of different molecular pathways [73]. While mitochondrial defects represent a precipitating factor for some genetic forms of PD, ALS, and Charcot-Marie Tooth disease, mitochondrial dysfunction also plays a very important role in the neuropathology of many neurodegenerative diseases (such as AD), being also considered one of the triggering factors of disease onset and progression [64,84]. Several studies have been conducted to improve HCI analysis that focus on mitochondrial phenotypic readouts and allow the evaluation of mitochondrial dynamics and homeostasis in iPSC-based neurodegenerative disease modelling (for a more detailed description, see [64]).

The works of the Zink group [33,34] provided a description of a novel high-throughput assay to quantify mitochondrial fitness and neuronal toxicity, specifically designed for neural cells obtained from human iPSCs, by performing a so-called mitochondrial neuronal health (MNH) assay. The assay combines live-cell measurement of mitochondrial membrane potential (MMP) with quantification of neuronal branching complexity (the latter according to the Lickfett et al., 2022 protocol [32], see Section 2.1). The protocol takes advantage of the potentiometric fluorescent dye tetramethylrhodamine methyl ester (TMRM) for evaluating (in live-imaging mode or in post-fixation assay) both the mitochondrial MMP and organelle morphology in the extending labelled neurites. HCI was performed throughout the HCA confocal microscope system with fitted automation imaging software analysis, while the analysis of the image dataset was suggested to be carried out by using two different pipelines: the first one is based on the open-source software CellProfiler, and the second pipeline is based on the image data storage and analysis system Columbus (PerkinElmer, Waltham, U.S.). Both of the imaging analysis setups are reported in detail in the work’s paper and also available with a relative tutorial in the provided web links [33,34].

A further example is provided by the work of Little et al. (2018), showing reduced MMP and altered mitochondrial morphology in PD neurons compared to control neurons [35]. Here, a HCI assay was used to simultaneously measure mitochondrial function, morphology, and cell viability in iPSC-derived dopaminergic neurons from PD patients with mutations in Synuclein Alpha (*SNCA)* gene and unaffected controls. The cells were stained with the MMP-dependent dye TMRM, alongside Hoechst-33342, and a calcein-based cell-permeant dye, used to determine cell viability (Calcein-AM). Images were acquired using an automated confocal screening microscope, and single cells were analyzed using the automated image analysis software CellProfiler (version 2.1.1). In this way, the authors demonstrated that PD neurons display reduced MMP and altered mitochondrial morphology compared to control neurons.

Therefore, these high-content methods can be successfully applied to the analysis of mitochondria in iPSC-derived neurons and can be exploited to test potential therapeutics for all the neurodegenerative diseases similarly characterized by mitochondrial dysfunctions.

## 4. HCI Analysis for Drug Screening and Neurotoxicity Assays in hiPSC-Based Neurodegenerative Disease Modelling

The above-described use and improvement of large-scale phenotyping approaches and HCI screenings for neuron morphology, long-range intracellular transport, protein aggregation, or altered organelle homeostasis using human iPSC-derived neurons have paved the way, in addition to the investigation of specific neuropathological mechanisms, for new drug screening strategies and precision medicine in neurodegenerative diseases. Indeed, the creation of integrated workflows for cell cultures, the numerous HCI methods, and consultable platforms for different degrees of data analysis automation (described above) have provided a powerful tool for drug screening studies specifically based on patient-derived hiPSCs as a complex investigation model that could precede human clinical trials for the treatment of neurodegenerative diseases [85].

### 4.1. Experimental Design and Analysis Settings

Many authors have developed protocols to perform drug screening on hiPSCs, potentially relevant for different pathologies.

The work of Sherman and Bang (2018) focused on HCI analysis of neurite growth, providing evidence that many pathways and targets known to play roles in neurite growth have similar activities in hiPSC-derived neurons that can be identified in an unbiased phenotypic screen [56]. Specifically, starting from a larger collection of bioactive small molecules (including FDA-approved drugs), the work identified 108 hit compounds able to regulate the neurite outgrowth. The specifically designed HCI assay for neurite outgrowth and inhibition was set up by using staurosporine (a broad-spectrum kinase inhibitor acting as a promoter of neurite outgrowth and branching [86]) as a positive control for establish analysis parameters for 384-well plated human iPSC-derived, cortical-like cell population screening. As previously reported for this type of phenotypic screening, the assay data were acquired on a confocal spinning disk microplate imaging system microscope, and all images were analyzed using the licensed software for automated analysis Columbus Acapella (PerkinElmer, Waltham, U.S.) to identify nuclei and perform neurite segmentation analysis [56]. Therefore, the work’s screening provided a powerful tool for phenotypic discrimination analyses of hiPSC-based disease models in patients in comparison to cells, highlighting disease-related molecular vulnerabilities.

HCI can be also useful to study specific molecular pathways and their pharmacological modulation. The work of Papandreou et al. (2023) reported a high-content screening of small molecule libraries in 11 patient-derived ventral midbrain progenitor cells in 96-well plates for an autophagy LC3-based assay. They studied how selected candidate small molecules restored or ameliorated the patient-specific phenotypic deficit [51]. The work further provided an ImageJ macros pipeline suitable for LC3 staining-based autophagosomes quantification (in this protocol compounds that significantly enhanced LC3 puncta production were indeed selected) and also shared technical tips to improve this type of drug screening approach in homogeneous samples of patient iPSCs [51]. Since autophagy is involved in the pathological change occurrence of many neurodegenerative disorders (including AD, PD, HD, ALS, and SMA), such a protocol/approach could be successfully applied in different research fields to screen efficient drugs.

Particular attention was also paid in setting up an HCI analysis workflow to screen and select potential therapeutic compounds that improve mitochondrial health, dynamics, and homeostasis in iPSC-based neurodegenerative disease modelling. The work of MacMullen and Davis (2021) provides a step-by-step protocol to perform rapid live-imaging mitochondrial phenotypic analysis of compounds with effects on mitochondrial dysregulations in differentiated hiPSC-derived neurons [64]. With this aim, hiPSCs differentiated into excitatory human neurons were further exposed to lentivirus vector infection to express a mitochondrial-targeted fluorescence reporter (TagGFP2 reporter), enabling morphological investigation of the neuronal organelles. Therefore, 384-well-plated glutamatergic neurons were treated with 4-hydroxychalcone and 2,4-dihyrdroxychalcone, belonging to a class of compound modulators of neuronal mitostasis (MnMs), able to influence mitochondrial features (number, elongation, and circularity) [63]. The phenotypic effects of the administered compounds were measured in neurons after live HCI by automated HCA confocal microscope image acquisition and, subsequently, automated analysis based on image segmentation in individual mitochondria (according to Varkuti et al., 2020 [63]) using the GE Developer Toolbox (1.9.2, build 2415, GE Healthcare, Chicago, U.S.) licensed software. In detail, the used segmentation workflow encompasses masking of somatic mitochondria, allowing measurements of axonal and dendritic mitochondria; then, the software generates data on the count of cell bodies and on the number, area, median circularity, and length of mitochondria. The authors demonstrated that MnMs had positive effects on iPSC-derived neurons by inducing morphological changes in mitochondria (reduced circularity and improved length) compared to controls. Therefore, the work supports the use of the iPSC-based HCI mitochondrial phenotypic assay for select key compounds effective on mitochondrial dysregulations linked to neurodegeneration.

### 4.2. Neuromuscular Diseases

HCI and drug screening approaches have been rather exploited in the field of neuromuscular diseases.

With the aim of evaluating in real-time live imaging neuromuscular junction (NMJ) formation, axonal outgrowth, and/or muscle maturation, the Osaki group set up an iPSC-derived on-chip 3D neuromuscular model [15]. The work reported the design and exploitation step process of a 3D physiological and pathophysiological motor unit model consisting of MNs coupled to skeletal muscles interacting via the NMJ within a microfluidic device. Of note, after acquiring the images (with a confocal microscope with a motorized stage and phase contrast and a fluorescent microscope with a stage-top incubator), the authors recommended performing automated image processing for high-throughput data collection by using different methods. For instance, they suggested the use of Python or Fiji programming (Image J) to (i) quantify microfluidic device pillar array deflection and (ii) measure contractile force muscles according to the area change and pillar edge tracking. While providing detailed instructions for both different analysis methods (FIJI mMacro for calculating pillar deflection is indicated), the protocols also suggested the alternative use of the Python/OpenCV package, enabling the automatic quantification of the displacement of the pillar and muscle contractile force from the captured image dataset [15]. In the field of neuromuscular diseases, the proposed system represents a good model to be applied in the pipeline of drug screening preceding clinical trials.

Other examples are provided by the works of Fujimori et al. (2018) and Rehbach et al. (2019), in which selected drug libraries were screened on several sporadic heterogeneous models of ALS iPSCs and HSP iPSC neurons, respectively [40,60]. In both works, the hit compounds were selected as potential therapeutic candidates (ropinirole for ALS models and a liver X receptor (LXR) agonist for HSP models) for their rescue of pathological neuronal morphology phenotypes, of which the measurements were evaluated according to the designed HCI analysis setup (described in Section 2.2).

### 4.3. Parkinson’s Disease

An example of HCI in PD modelling has been provided in the work of Antoniou et al. (2022), in which HCA was performed on iPSC-derived neurons from patients with familial PD [bearing the G209A (p.A53T) α-synuclein (αSyn) mutation] for the evaluation of disease-associated neurodegenerative phenotypes being rescued by a multi-kinase inhibitor BX795 [26]. Since the therapeutic targets of PD are mainly represented by the aggregation of misfolded αSyn as the main pathogenic factor causing cellular toxicity, the work aimed to evaluate the neuroprotective phenotypic effect of BX795, an inhibitor of 3-phosphoinositide-dependent kinase 1 (PDPK1). The compound, identified from a screening campaign on a small kinase inhibitor library, was tested for its phenotypic effects by quantifying the degree of impaired neuritic outgrowth, dystrophic neurites or fragmented cells, the presence of intracellular protein aggregates, and maintenance of axonal integrity in treated neurons, compared to untreated cells. Furthermore, the analysis was performed at both early and later stages of neuronal maturation, when disease-associated phenotypes were already established. Images were captured by automated HCA-confocal microscopy, while image segmentation parameters were set as follows: primary object detection (cell nuclei) was based on Hoechst staining; detection of neurons was based on TUJ1 immunofluorescence signals and on TH immunofluorescence signals. For quantification of TUJ1 and TH intensity, the authors used Image Mining, a custom-made image processing and analysis application with an extendable “plug-in” infrastructure [44]. Neurite length was estimated manually by tracing the length of all neurites on TH-labeled neurons using the NeuronJ plug-in of ImageJ. The number of TUJ1+ spots in blebbed or fragmented axons was counted manually (ImageJ), and the ratio between the number of spots and the total TUJ1+ staining area (ImageJ) was defined as the axon degeneration index [41,42,43]. The work results, also reinforced by proteomic analysis outcomes, highlighted the neuroprotective effects of multi-kinase inhibitor BX795 on PD-iPSCs neurons, showing the rescue of disease-associated neurodegenerative phenotypes. Providing a promising novel small molecule as a candidate therapeutic for PD and other protein conformational disorders, the work also represents a successful example of combining high-throughput screening approaches iPSC-based disease modelling as a promising unbiased strategy to identify therapies for neurodegenerative disorders.

### 4.4. Alzheimer’s Disease

Moreover, in the AD field, several studies have exploited HCI analysis to obtain in-depth phenotypic characterizations of neurite dysmorphogenesis associated with aberrant accumulation of Aβ and p-Tau, thus providing new foundations for drug screening applications.

For instance, the HCI screening of a Library of Pharmacologically Active Compounds (LOPAC) in AD iPSC-derived neurons allowed Wang and colleagues (2017) [65] to identify moxonidine and metaproterenol (two adrenergic receptor agonists) as endogenous human tau-lowering compounds. In detail, 384-well-plated differentiated glutamatergic neurons from an integrated, inducible, and isogenic Ngn2 iPSC line (i3N) were treated with 1280 compounds from the LOPAC library, and then human tau, total neurites, and nuclei were detected by an immunostaining procedure and images acquired with the fully automated ArrayScan high-content system (Thermo Fisher Scientific, Waltham, U.S). Tau-lowering hits were selected according to automated quantification of multiple cellular and well neuronal features with Cellomics software (Thermo Fisher Scientific, Waltham, U.S), quantification of viable cell number (based on nuclei count and characterization), neurite total length, and total tau levels. The described approach represents a reliable tool for drug screening-based research for AD and tauopathies therapies [65].

Other examples come from the works of Chang et al. (2019) and Bassil et al. (2021) (already mentioned in the present review [see Section 2.2]), which provided HCI analysis of neurite outgrowth in AD- and wild-type iPSC-derived neurons, together with aberrant accumulation of Aβ and tau phosphorylation [57,58]. By establishing an HCI analysis workflow for the neurite outgrowth measurements of differentiated neurons, Chang et al. performed a small-molecule screening on AD-iPSC-derived neurons carrying the APP-D678H mutation. The HCI analysis suggested the neuroprotective effect of the synthetic indolylquinoline compound NC009-1, since it was able to improve neuronal cell viability and neurite outgrowth [57]. In a similar way, Bassil et al. (2021) performed the screening of a library of 70 small molecules and natural products and investigated by HCI analysis in the established culturing platform of human iPSC-AD neurons, astrocytes, and microglia [see Section 2.2] the neuroprotective effect of anti-Aβ antibody treatments. Based on the area quantification of MAP2+ dendrites and Aβ plaque (X04 stained), synapse count (synapsin labeled), and p-Tau 396–404 induction fold (S235 stained), the work demonstrated that the anti-Aβ intervention was able to slow down neuron degeneration and plaque formation in a precise therapeutic time window. Therefore, the study highlighted the value of investigating the AD model high-throughput culture platform used with HCI analysis to screen potential therapeutic compounds by also defining the intervention window and deepening our understanding of AD [58].

### 4.5. Neurotoxicity Assays

HCI approaches can also be used to perform neurotoxicity assays on hiPSC-derived neurons.

Indeed, tools have been optimized to measure synaptic density changes (triggering cell death pathways). For instance, the work of Green et al. designed an automated synaptic imaging assay (ASIA) that allows for the labelling and imaging of live neuron synapses with both confocal and wide-field microscopy (the latter requires a deconvolution step in the analysis) [36]. To this aim, the work exploits the use of viral transduction-induced fluorescent protein expression (PSD95-eGFP) to identify agents that regulate synapse number in iPSC-derived cortical neurons exposed to different doses of glutamate. The automated image acquisition protocol, applied to a confocal microscope controlled with the JOBS module for design of automated acquisition (NIS-Elements software, Nikon, Tokyo, Japan) and possibly equipped with a Photometrics CMOS camera and a LED light source for wide-field imaging, was integrated into JOBS to define a plate alignment function that allows automatic imaging from the same ROI over time. The work then shared the basic workflow image analysis protocols (with related description of sequential steps for image processing and the files for running the analysis) to obtain the primary analysis endpoints (the number of structural synapses and cell viability) in the open-source software platforms MetaMorph, ImageJ, and CellProfiler [36]. Therefore, ASIA is an efficient approach to label, image, and analyze synapses in live neurons: it can help in identifying agents that evoke synaptic toxicities and screening for compounds preventing or reversing synapse loss.

Additionally, HCI methods in hiPSCs have also been employed to combine quantitative epidemiological study and assessment of neurodegenerative hallmarks with compound toxicity screening in patient neurons. In this regard, a recent work of Paul et al. (2023) reported effects of pesticide exposures in dopaminergic neurons derived from PD hiPSCs to identify Parkinson’s-relevant pesticides [87]. In detail, selecting a subclass of compounds in a library of specific pesticides associated with PD risk in a comprehensive pesticide-wide association study, a live-cell imaging screening was performed in the work for patient-derived dopaminergic neurons exposed to a large number of pesticides (possible used in combinations in cotton farming) throughout a chamber-equipped high-throughput microscope. Cultured iPSCs were previously engineered with THtdTomato fluorescent reporter, since endogenous THtdTomato signal colocalized with anti-TH-labelling enabled selective evaluation of the effects of pesticides on midbrain dopaminergic differentiated neurons and excluded other contaminating cell types present in patterned iPSC-derived cultures. Therefore, a live-cell images dataset was analyzed using the licensed software Columbus (PerkinElmer, Waltham, U.S.) for automated imaging analysis that enables the segmentation algorithm of fluorescently labelled nuclei and neurites. The analysis was in fact performed in several phases: the first aimed at identifying the live cells that met size and roundness criteria while excluding the debris; in the second phase, neurite detection algorithms were applied in order to identify neurites based on the THtdTomato signal in these cellular processes using the Find Neurites, CSIRO neurite analysis method (setting parameters are available in paper methods) [87].

## 5. Conclusions

The present review aimed to provide useful suggestions to guide the choice and optimization of study designs for the use of hiPSC-derived neurons in neurodegenerative diseases modelling by exploiting HCI methods. These approaches enable the achievement of high-quality in vitro data for the characterization of new disease pathways, precision medicine, and drug screening. Therefore, reporting the most recent improvements in experimental designs published in last years, we dissected iPSC-plating methods, tools of fluorescent labelling for organelle tracking, and, most importantly, microscope requirements, imaging platforms, data analytics, and management tools to be employed. All the reported works shed light on the currently achievable applications in iPSC-based HCI disease modelling and HCI, both to recapitulate/predict disease phenotypes in vivo and to gain new insights into the heterogeneity of disease manifestations, thereby improving drug screening applications. The limitations of all presented protocols are based on their foundation with two-dimensional (2D) cell cultures. Research based on three-dimensional (3D) models (spheroids, organoids, and assembloids) is rapidly growing due to the improvement of morphological characteristics, cellular complexity, and the physiological relevance of cell systems, which appear to be better suited for drug discovery applications than 2D monolayer cultures [88,89,90,91]. However, the transition from 2D monolayer models to 3D spheroids in HCI applications is challenging and requires 3D-optimized protocols, instrumentation, and resources. Thus, while emerging 3D technologies and platforms aim to overcome such limitations for the successful implementation of HCI assays in 3D models, HCI as single cells in patient-derived iPSCs, leveraging all recent improvements, represents an accessible and powerful tool for modelling and addressing neurodegenerative disease research issues.

## Figures and Tables

**Figure 1 ijms-24-14689-f001:**
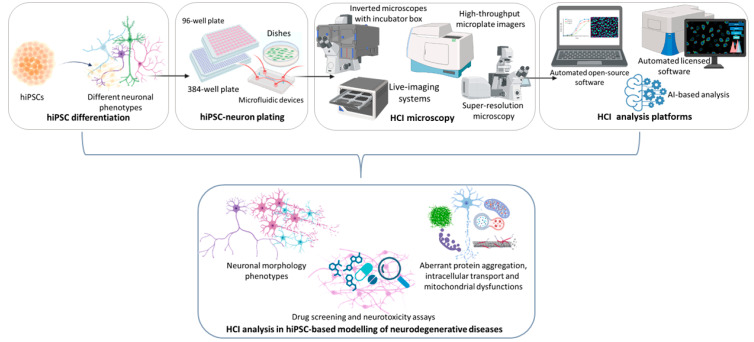
Workflow for HCI and analysis in hiPSC-based modeling of neurodegenerative diseases. Created with BioRender software (BioRender.com) (accessed on 24 September 2023).

**Figure 2 ijms-24-14689-f002:**
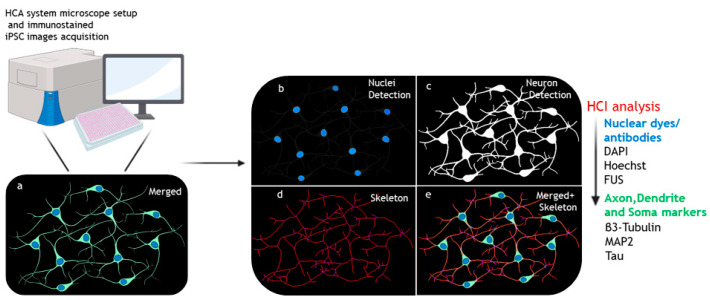
HCI for neuronal morphology phenotypes. Analysis workflow for automated skeletonization of neurite outgrowth. Representative cartoon for basic analysis pipeline for neurite outgrowth quantification. Fixed iPSC-neurons are automated and analyzed by software that enables image segmentation, breaking down the image into discrete objects (such as single nuclei or neurites), detected according to immunostaining of specific markers (most commonly used are indicated on the right). Typically, the nuclei staining (reference input for cell counting) (**a**,**b**) and neuronal staining channel (**a**) are detected, and then the binary images for neurites and soma area (**c**) are processed. The neurite skeleton is produced (**d**) and evaluated (after soma subtraction) (**e**) for neurite outgrowth analysis. Created with BioRender software (BioRender.com) (accessed on 24 September 2023).

**Figure 3 ijms-24-14689-f003:**
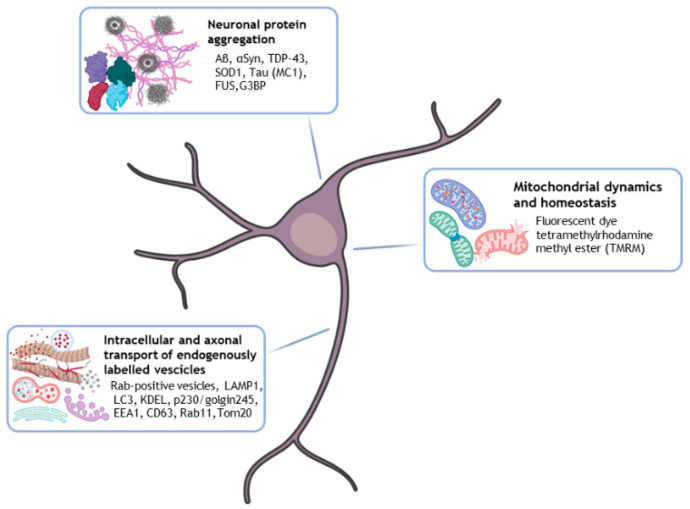
HCI for studying (i) aberrant neuronal protein aggregation, (ii) intracellular and axonal transport of endogenously labelled vesicles, and (iii) mitochondrial dynamics and homeostasis in iPSC-based neurodegenerative disease models. In culture or fixed iPSC-derived neurons can be automatically analyzed by HCI software, which enables image segmentation and identification of different neurodegeneration-related molecular pathways (the most commonly used markers are listed in the figure captions). Created with BioRender software (BioRender.com) (accessed on 24 September 2023).

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
