# Peer review of "Recent Advances in High-Content Imaging and Analysis in iPSC-Based Modelling of Neurodegenerative Diseases"

_ijms, 2023, doi:10.3390/ijms241914689_

Round 1
Reviewer 1 Report
In this review, Menduti et al. have summarized the recent advances in high-content imaging (HCI) analysis in iPSC-based neurodegenerative disease modeling. The topic is highly interesting and relevant and the manuscript is comprehensive and well written. There are a few suggestions:
1. Some of the sentences are very long and confusing and some have typos and are grammatically incorrect. It is recommended that the authors rephrase/edit these. Some examples include:
A. "In particular the neuronal morphology HCI analysis were obtained in 96 well-plates fixed neurons throughout the Harmony software (PerkinElmer), a licensed HCI analysis software providing “building blocks” for image segmentation analysis, of which pipeline involved breaking down the image into discrete objects (such as individual nuclei or neurites), detected according immunostaining of specific markers. "
B. "The experimental evaluations have been carried out on low Aβ-secreting neuroprogenitor cells and high Aβ-secreting mature neurons hiPSC-derived neurons, in which flAPP/BACE1 colocalization and APP-derived C-terminal fragment and presenilin-1 (the catalytic subunit of γ-secretase) colocalization were evaluated with immunocytochemistry combined with a proximity ligation assay (PLA). "
C. "Since the therapeutic targets of PD are mainly represented by the aggregation of misfolded αSyn as the main pathogenic factor causing cellular toxicity, the analysis of the work aimed to evaluate the neuroprotective effect of BX795, identified from a screening campaign on a small kinase inhibitor library, by quantifying the degree of impaired neuritic outgrowth, dystrophic neurites or fragmented cells, the presence of intracellular protein aggregates and maintenance of axonal integrity in treated neurons compared to untreated ones, both in the early neuronal differentiation or at later stages of neuronal maturation, when disease-associated phenotypes were already established. "
2. In the introduction, the authors state "Recent HCI analysis of iPSC-derived neurons have led to enabling in-depth studies of several hallmarks related to neurodegeneration, classifiable to: i) neuronal dysmorphogenesis and survival; ii) aberrant neuronal protein aggregation and intracellular transport; ii) compound-induced neurotoxicity". The authors should include Mitochondrial dysfunction as the third bullet point. This should also be depicted in Figure 1.
3. The authors refer to the work by Booker et al. and mention that "The work provides a robust approach for live imaging and measurement of axonal and dendritic transport in iPSC-derived cortical neurons by exploiting transient overex- pression of organelle markers, and generating LAMP1-EGFP knock-in iPSCs to study transport of fluorescently labelled or endogenously labelled vesicles, respectively. " How were the axonal projections distinguished from the dendrites in this study?
4. References [42] and [43] are same and stated twice.
5. In Section 4. the authors should include how HCI using iPSC derived neurons can be used to identify compounds that influence mitochondrial health (As described in MacMullen et al., 2021).
6. All the abbreviations used in the text should be expanded (For example: MND etc.).
The text is well written in sound English. However there are typos and some sentences are severely long (as mentioned in the major comments).
Author Response
Thank you so much for taking the time to review this manuscript. Please see the attachment with the detailed responses and corresponding revisions/corrections highlighted in the resubmitted files.

Reviewer 2 Report
The present review represents an accurate revision of the recent improvements to analyze experimental designs about human iPSC derived models of human neurodegeneration diseases, by high-throughput screening with subcellular resolution microscopy for cell-based high-content imaging (HCI) imaging platforms, together with data analytics and management tools to be employed. The revision describes and discuses iPSC plating methods, tools of fluorescent labelling for organelle tracking and microscope requirements. It is therefore an useful guide the choice and optimization of study designs for the use of hiPSC-derived neurons in neurodegenerative dis-eases modelling by exploiting HCI methods
I consider the present version in clear and informative and ready to be suitable for publication.
Author Response

(The authors gave the same response as above.)

Reviewer 3 Report
Summary:
The review titled "High-content imaging and analysis in iPSC-based modelling of neurodegenerative diseases: recent advances" by Giovanna Menduti and Marina Boido delves into the advancements in the field of neurodegenerative pathologies using patient-derived induced pluripotent stem cells (iPSCs). These platforms have emerged as significant molecular diagnostic/prognostic tools, enabling in vitro understanding of neurodegenerative mechanisms and molecular heterogeneity of disease manifestations. The review emphasizes the integration of high-throughput screening with subcellular resolution microscopy for high-content imaging (HCI) in iPSC-derived neuronal cells. This combination facilitates comprehensive analyses of cell morphology and neurite trafficking using state-of-the-art microscopes and automated computational assays. The authors aim to outline the latest protocols and achievements in HCI analysis for iPSC-based neurodegenerative disease modelling, emphasizing technical and bioinformatics strategies for future research.
Comments:
The review is comprehensive and provides a detailed overview of the advancements in iPSC-based modelling for neurodegenerative diseases using HCI. The abstract is well-structured, providing a clear understanding of the topic's significance and the review's objectives.
Major points:
1. Title: The title "High-content imaging and analysis in iPSC-based modelling of neurodegenerative diseases: recent advances" is descriptive and encompasses the core focus of the review. However, it might benefit from a slight modification to “Recent Advances in High-Content Imaging and Analysis for iPSC-Based Neurodegenerative Disease Modeling”.
2. Key words should include” iPSC” to more precisely with the content of this review.
3. Reference: The review exhibits a noticeable deficiency in references, with several sections, including the Introduction, lacking the necessary citations to support the provided information. For instance, there is a missing reference immediately after 'As first described by Yamanaka et al. in 2006,' which leaves a significant gap in the literature coverage.
4. Subtitles: i) 1.1 should be correct as of “Most Common Microscope Settings and Platform Analysis in iPSC-Based Neuronal Models”
5. More figures are needed:
It is recommended to include more figures throughout the paper to visually illustrate and clarify the authors' key points. Visual aids can significantly enhance the readers' understanding of the discussed concepts.
6. Inclusion of Alzheimer's Disease Section:
Please consider adding a dedicated section to discuss Alzheimer's disease (AD), as it holds immense significance within the realm of neurodegenerative diseases. Addressing AD in a distinct section would provide valuable insights into this prominent condition and its relevance to the broader topic of neurodegenerative diseases.
No big issue is found in this manuscript.
Author Response

(The authors gave the same response as above.)

Round 2
Reviewer 3 Report
These authors have addressed my concern in their revised manuscript.